# Establishment of Immune Biobank for Vaccine Immunogenicity Prediction Using In Vitro and In Silico Methods Against Porcine Reproductive and Respiratory Syndrome Virus

**DOI:** 10.3390/vaccines13101052

**Published:** 2025-10-14

**Authors:** Chaitawat Sirisereewan, John J. Byrne, Lanre Sulaiman, Abigail Williams, Ben M. Hause, Juliana Bonin Ferreira, Glen W. Almond, Benjamin Gabriel, Anne S. De Groot, Tobias Käser, Gustavo Machado, Elisa Crisci

**Affiliations:** 1Department of Population Health and Pathobiology, College of Veterinary Medicine, North Carolina State University, Raleigh, NC 27607, USA; csirise@ncsu.edu (C.S.); jakejbyrne1@gmail.com (J.J.B.); laprecieux@gmail.com (L.S.); aewill25@ncsu.edu (A.W.); jboninf@ncsu.edu (J.B.F.); gwalmond@ncsu.edu (G.W.A.); tobias.kaeser@vetmeduni.ac.at (T.K.); gmachad@ncsu.edu (G.M.); 2Cambridge Technologies, Worthington, MN 56187, USA; bhause@cambridgetechnologies.com; 3EpiVax Inc., Providence, RI 02909, USA; bgabriel@epivax.com (B.G.); annied@epivax.com (A.S.D.G.); 4Department of Biomedical Sciences and Pathobiology, Institute of Immunology, University of Veterinary Medicine Vienna, 1210 Vienna, Austria

**Keywords:** PRRSV, immune bank, vaccine immunogenicity, immune responses, EpiCC, T cell epitope

## Abstract

Background/Objectives: Porcine reproductive and respiratory syndrome virus (PRRSV) remains one of the most economically significant pathogens in the global swine industry. Despite the availability of commercial vaccines for over three decades, they fail to induce sterile immunity and often provide inconsistent protection against heterologous PRRSV strains. This study aimed to predict vaccine immunogenicity by detecting strain-specific immune responses that related to an immune correlate of protection (CoP) against different PRRSV-2 strains. Methods: Post-weaning pigs were vaccinated with five commercially available PRRSV-2 vaccines or received sterile PBS injection as a control. At 28 days post-vaccination (dpv), all pigs were humanely euthanized for large-volume blood collection to isolate peripheral blood mononuclear cells (PBMCs) and plasma, establishing the immune bank. PBMCs and plasma from each group were then tested against six PRRSV-2 strains to evaluate immune responses. In addition, T cell epitope coverage between vaccine and field PRRSV-2 strains was assessed using the EpiCC (in silico) tool to enhance predictive capacity. Results: While neutralizing antibodies were undetectable in all vaccinated pigs at 28 dpv, PRRSV-specific IFNγ–producing cells were detected at various levels in each vaccinated group following restimulation with different PRRSV-2 strains. Additionally, a positive correlation was observed for the EpiCC coverage of the N gene and mean IFNγ responses to VR2332 (SLA class I and II) and NC24-6 (SLA class II). Conclusions: The PRRSV immune bank demonstrated potential as a tool for predicting vaccine immunogenicity against different PRRSV-2 strains and EpiCC provides additional information on T cell epitope cross conservation. The combined approach may provide a valuable framework for selecting PRRSV vaccines for more effective prevention and control in endemic areas.

## 1. Introduction

Since its first identification, porcine reproductive and respiratory syndrome virus (PRRSV) has remained one of the most problematic swine viruses across the globe. Notably, the economic impact of PRRSV infection rose to $1.2 billion annually between 2016 and 2020 in US swine industry [1], marking a sharp increase compared to the previous decade. Pigs infected with PRRSV exhibit delayed innate and adaptive immune responses due to the virus’s immunomodulatory effects, which contribute to increased susceptibility to secondary infections, known as porcine respiratory disease complex (PRDC) [2,3,4,5]. Moreover, the continuous evolution and emergence of new strains, driven by a high mutation rate and frequent recombination events [6,7], pose a major challenge to effective control and eradication of PRRSV. Commercial PRRS modified-live virus vaccines remain the primary tools for controlling PRRSV and stabilizing herds for more than three decades [8,9]. However, the immunity induced by these vaccines has shown limited effectiveness in protecting against heterologous strains and newly emerging variants [10,11,12,13], and can result in immune dysregulation comparable to that caused by wild-type strains [14,15,16]. Thus, the development of effective vaccines capable of conferring sterile immunity across heterologous strains is essential; however, such vaccines are not expected to be available soon. This limitation highlights a critical question: which vaccine strains are most effective in mitigating outbreaks caused by newly emerging PRRSV strains?

This study aims to assess vaccine immunogenicity, including both humoral and cell-mediated responses, to identify commercial PRRSV-2 vaccines that elicit strong protection against various PRRSV-2 strains. Neutralizing antibodies (nAbs) are crucial for viral clearance and guarding against reinfection [17,18]. However, vaccine-induced neutralizing antibodies often develop slowly, are produced inconsistently, with limited cross-reactivity across different PRRSV strains [19,20]. Furthermore, PRRSV can persist in infected tissues even in the presence of neutralizing antibodies [21,22]. These factors suggest that cell-mediated immune responses likely play a more crucial role in viral clearance and in controlling viral infection, both before and after the emergence of nAbs [23].

To properly select PRRSV vaccines to protect pigs from PRRSV outbreaks, we developed a biobank of PRRSV-vaccine exposed immune cells and an approach called Predict and Protect against PRRSV (PreProPRRSV), to predict vaccine immunogenicity. In this study, vaccine immunogenicity was assessed by measuring strain-specific immune responses after in vitro restimulation with circulating PRRSV-2 strains from the North Carolina (NC) swine industry, which have been identified as strain-specific immune correlates of protection (CoP) based on findings from previous studies [19,24]. CoP was defined as the measurable immunological characteristics and mechanisms associated with protection from disease and responsible for vaccine-induced efficacy [25,26,27]. In this study, we focused on two key arms of the CoP against PRRSV: nAbs, which serve as a humoral shield to prevent infection, and cell-mediated immunity, which is critical for broad and effective viral clearance. Both have been identified as relevant CoP in the PRRSV literature [19]. In parallel, we utilized the Epitope Content Comparison (EpiCC) algorithm to assess the potential contribution of T cell epitope cross-conservation between vaccine strains and field isolates to protective immunity [28,29]. The findings of this study offer valuable insights for predicting vaccine immunogenicity and guiding vaccine selection against newly emerging PRRSV strains in endemic regions.

## 2. Materials and Methods

### 2.1. Experimental Designs

Thirty-six, 3-week-old commercial breed pigs were purchased from a PRRS-negative farm and transported to the BSL-2 Laboratory Animal Research (LAR) facility at NC State University’s College of Veterinary Medicine (Raleigh, NC, USA). Upon arrival, the pigs were ear-tagged for identification and acclimated for 7 days. Prior to vaccination, the PRRSV-negative status was reconfirmed using a commercial ELISA (IDEXX PRRS X3 antibody test, Westbrook, ME, USA) and PRRSV-specific RT-qPCR assay. To establish the biobank, pigs were randomly assigned to six groups, with groups balanced for weight and sex, including the control group (Group 1) and five vaccinated groups (Groups 2–6), as detailed in Figure 1. At 4 weeks of age, pigs in the control group (*n* = 6) were injected with sterile PBS as a negative control, while pigs in each vaccinated group (*n* = 6) received an intramuscular injection of one of the following commercial PRRSV vaccines: Fostera, Ingelvac, PrimePac, Prevacent, and PRRSGard, according to the manufacturer’s instructions. All groups were humanely euthanized 28 days post vaccination (dpv) to collect plasma and peripheral blood mononuclear cells (PBMCs) for immune biobank establishment. Moreover, average daily gain (ADG) at 28 dpv was assessed in vaccinated and control pigs and is presented in Appendix A. All methods and animal studies were conducted under the approval of NC State University Animal Care and Use Committee (IACUC) ID# 20-468, 23-369.

### 2.2. Viruses and Cells

The circulating and newly emerging NC PRRSV-2 strains used in this study (Appendix A) were isolated and propagated in both the MA104 cell line (ATCC, CRL-2378.1T) and porcine alveolar macrophages (PAMs). The reference strain, ATCC^®^ VR-2332™ (GenBank: U87392), was propagated in MA104 cell line. After successful isolation, the viruses were submitted for sequencing by an outsourced sequencing service and characterized according to the fine-scale classification of PRRSV-2, as previously described [30,31]. These viruses were subsequently used in immunological assays to investigate both humoral and cellular immune responses following vaccination. Virus titers were determined using the immunoperoxidase monolayer assay (IPMA) and expressed as TCID_50_/mL, determined by the Reed–Muench method [32].

### 2.3. Immunoperoxidase Monolayer Assay (IPMA)

IPMA was performed for the detection the PRRSV antigen. Briefly, PRRSV infected cells were fixed with a 1:1 methanol-acetone solution for 15 min at room temperature (RT), then washed twice with phosphate-buffered saline containing 0.5% Tween-20 (0.5% PBST). The cells were stained with an anti-PRRSV N protein monoclonal antibody (SR30-A, Brooking, SD, USA) diluted 1:1000 and incubated for 60 min at RT. After two washes with 0.5% PBST, the cells were incubated with HRP-conjugated rabbit anti-mouse antibody (ab6728, Abcam, Waltham, MA, USA), diluted 1:1000, for 60 min at RT. Following two additional washes with 0.5% PBST, the cells were counterstained with 3-amino-9-ethylcarbazole (AEC) substrate. The presence of PRRSV antigens was examined under a microscope and titers were subsequently calculated.

### 2.4. Quantification of PRRSV RNA

RNA was extracted from plasma and virus-infected cells using the PureLink Viral RNA/DNA Mini Kit (Invitrogen, Waltham, MA, USA) according to the manufacturer’s instructions. The eluted RNA was quantified using a Nanodrop and then converted to cDNA using Applied Biosystems TaqMan^®^ Reverse Transcription (Applied Biosystems, Waltham, MA, USA). The cDNA was subsequently used for quantitative PCR (qPCR) with IQ SYBR^®^ Green Supermix (Bio-RAD, Hercules, CA, USA). PRRSV RNA quantification was performed as previously described [33] with minor modifications, utilizing SYBR green-based real-time qPCR.

### 2.5. Phylogenetic Analyses

All nucleotide and amino acid sequences of NC PRRSV-2 strains were aligned with the vaccine strains used in this study using the Clustal W algorithm in BioEdit version 7.2.5 (https://bioedit.software.informer.com/, accessed on 10 October 2025, Raleigh, NC, USA). Phylogenetic trees were constructed in MEGA version 11 using the Maximum Likelihood method with 1000 bootstrap replicates. The GTR+G substitution model was applied for nucleotide sequences, and the JTT+G model was used for amino acid sequences [34].

### 2.6. Serological Assays

PRRSV-specific antibody responses were measured using the PRRSV X3 enzyme-linked immunosorbent assay (ELISA, IDEXX, Westbrook, ME, USA) both prior to vaccination and 28 dpv. Samples were considered positive if the *S*/*P* ratio exceeded 0.4, according to manufacturer’s instructions.

Additionally, virus neutralization (VN) assay against the several PRRSV-2 strains was performed using the previously described [35], with minor modifications, using 200 TCID_50_/50 µL of the viruses. A VN titer was considered positive at ≥1:2 (1 log_2_). PRRSV antigens were detected using the IPMA, examined microscopically, and VN titers were subsequently calculated.

### 2.7. Isolation of PBMCs

Heparinized blood samples were centrifuged at 1000× *g* for 20 min at 25 °C to separate plasma and buffy coat. The plasma was transferred to a 50 mL centrifuge tube, aliquoted into 1 mL portions, and stored at −80 °C. PBMCs were isolated from buffy coat diluted with 1× PBS by gradient centrifugation using SepMate tubes (StemCell, Vancouver, BC, Canada) and Ficoll-Paque (GE Healthcare, Uppsala, Sweden). After isolation, the cells were resuspended in RPMI 1640 medium (GIBCO, Carlsbad, CA, USA), supplemented with 10% heat-inactivated, fetal bovine serum (FBS), (GIBCO), 100 U/mL of penicillin G, 100 μg/mL of streptomycin, 0.01 mg/mL of gentamycin (GIBCO), and 2 mM L-glutamine (GIBCO) referred as complete media. The cell count was determined using Luna^®^ Automated counter (Logos biosystems, Annandale, VA, USA). The cells were then recentrifuged, resuspended in freezing media consisting of 60% FBS, 30% complete media, and 10% dimethyl sulfoxide (DMSO), aliquoted into cryotubes, and stored in liquid nitrogen until needed.

### 2.8. Enzyme-Linked Immunospot (ELISPOT) Assay

Briefly, 96-well filter plates with hydrophobic PVDF membranes (Millipore, Burlington, MA, USA) were treated with 35% ethanol for 1 min. After activation, the plates were washed five times with ddH_2_O and coated overnight at 4 °C with a monoclonal antibody specific to porcine interferon gamma (IFNγ) (1:50; Mabtech, Nacka Strand, Sweden). The antibody-coated plates were washed with PBS and incubated with stimulants, including the positive control (Concanavalin A, 5 µg/mL), NC PRRSV strains, and a reference PRRSV strain, respectively, at an MOI of 2. The wells containing only cells and complete media served as mock controls.

PBMCs were thawed, washed with RPMI-1640 medium, counted using trypan blue, and seeded at 2.5–5 × 10^5^ cells/well. The plates were incubated with stimulants at 37 °C in 5% CO_2_ overnight, then washed with PBS and incubated with a biotinylated IFNγ-specific antibody (clone P2C11, Mabtech, Nacka Strand, Sweden) at 1 µg/mL for 1 h at RT. After washing, streptavidin-alkaline phosphatase (Mabtech, Cat#3310-10-1000) was added at a 1:2000 dilution and incubated for 60 min at RT. The alkaline phosphatase substrate, 5-bromo-4-chloro-3-indolyl phosphate/nitro blue tetrazolium (BCIP/NBT; Sigma-Aldrich, St. Louis, MO, USA), was incubated for 25 min. The reaction was stopped by rinsing with tap water, and the plates were left to dry overnight.

Spot counting was performed using the Mabtech ELISpot reader (Mabtech, Nacka Strand, Sweden). PRRSV-specific IFNγ producing cells were expressed as spot-forming units (SFU) per million PBMCs in each well.

### 2.9. Prediction of T Cell Epitopes Using EpiCC Algorithm

The T cell epitopes predictions for both SLA class I and class II alleles were performed as previously described [28,29]. Briefly, amino acid sequences of the vaccine strains and six PRRSV field strains used in this study were analyzed to identify and assess the conservation of individual T cell epitopes across strains, in silico, yielding EpiCC scores that were subsequently converted to T cell epitope coverage. For each strain, eight structural proteins (GP2a, GP2b, GP3, GP4, GP5, GP5a, M, and N) were evaluated in the context of nine SLA Class II alleles (DRB1*0101, DRB1*0201, DRB1*0401, DRB1*0402, DRB1*0501, DRB1*0601, DRB1*0602, DRB1*0701, DRB1*1001) and 15 SLA Class I alleles (1*0101, 1*0401, 1*0701, 1*0801, 1*1201, 1*1301, 2*0101, 2*0401, 2*0501, 2*1001, 2*1201, 3*0401, 3*0501, 3*0601, 3*0701). Predictions specific to the SLA type of the pigs studied were not possible as the SLA type of the pigs used for these studies was not available. The EpiCC scores were calculated for predicted T cell epitopes as compared to the vaccine strain, resulting in a coverage score that represented the percentage of T cell epitope conserved between the strains. This coverage score is intended to represent potential cross-protection. The set of SLA alleles listed above were used in a previous study of PRRSV vaccine efficacy and immunogenicity for which a correlation between EpiCC coverage and protective efficacy was observed. The set of SLA alleles listed above were used in a previous study of PRRSV vaccine efficacy for which a correlation between EpiCC coverage and protective efficacy was observed [28].

### 2.10. Statistical Analyses

All analyses were performed using GraphPad Prism version 10 for Mac (GraphPad Software Inc., San Diego, CA, USA). Data were first tested for normality using the Shapiro–Wilk test to determine the appropriate statistical approach. For data sets that met the assumptions of normality, parametric tests were used, including analysis of variance (ANOVA) followed by Tukey’s multiple comparisons test. For non-normally distributed data, the Kruskal–Wallis test was applied, followed by uncorrected Dunn’s multiple comparisons test.

Correlation analyses were performed using a correlation matrix with Pearson correlation coefficients and a two-tailed 95% confidence interval. Nonlinear regression curves were fitted using a lognormal equation.

## 3. Results

### 3.1. Nucleotide and Amino Acid Similarity Based on Complete Genome Sequences Between Vaccine Strains and NC PRRSV-2 Strains

In this study, complete genome nucleotide and amino acid sequence similarities were analyzed between vaccine strains and NC PRRSV-2 field strains. Most vaccines showed high similarity to VR2332 at both levels, especially the Ingelvac MLV vaccine. When compared to NC PRRSV-2 strains, the vaccine strains shared nucleotide similarities ranging from 80.21% to 83.44% and amino acid similarities ranging from 52.93% to 60.48%. The phylogenetic trees based on complete genome and amino acid sequences of vaccine and different PRRSV-2 strains were shown in Figure 2A,B. The nucleotide and amino acid similarities based on complete genome sequence between vaccine strains and NC PRRSV-2 strains were summarized in Appendix A.

### 3.2. Humoral Immune Responses Following Vaccination

Prior to vaccination, all pigs were confirmed PRRSV-negative by both ELISA and qPCR, showing no seropositivity or viremia. At 28 dpv, all vaccinated groups exhibited elevated mean *S*/*P* ratios above the cut-off value of 0.4, indicating seroconversion. All vaccinated groups showed significant PRRSV-specific immune responses compared to the control group (Figure 3). Moreover, significant difference in immune responses were observed among vaccinated groups. These findings suggest that all commercial MLV vaccines evaluated in this study were capable of inducing humoral immune responses, albeit to varying degrees.

To determine whether PRRSV-specific antibodies were associated with protection, virus neutralization assay (VN) was performed against heterologous NC PRRSV-2 and reference strains. The results revealed none of the vaccinated pigs exhibited the virus neutralizing antibodies against any of the NC PRRSV-2 and reference strains at 28 dpv. Moreover, no correlation was observed between the mean *S*/*P* ratios and VN titers in any of the vaccinated groups.

### 3.3. Cell-Mediated Immune Responses Following Restimulation with NC PRRSV-2 Strains

At 28 dpv, PRRSV-specific IFNγ–producing cells were detected following heterologous restimulation with NC PRRSV-2 strains. The vaccinated pigs exhibited distinct profiles of IFNγ responses against both the NC field strains and the reference strain. For VR2332, pigs vaccinated with Fostera and Ingelvac vaccines showed significantly higher numbers of IFNγ–producing cells compared to the control and PRRSGard groups (Figure 4A). Upon NC134 restimulation, the PrimePac group exhibited significantly higher responses than the control, Fostera, and Prevacent groups (Figure 4B). For NC24-6, pigs in the Fostera, Ingelvac, and PrimePac groups had significantly higher IFNγ–producing cell levels than those in the control and PRRSGard groups (Figure 4C). Following NC20-1 restimulation, the PrimePac group also showed significantly elevated IFNγ responses compared to the control and Ingelvac groups (Figure 4D). After NC24-9 restimulation, all vaccinated groups except PRRSGard exhibited significantly higher IFNγ responses compared to the control (Figure 4E). Additionally, the Ingelvac group demonstrated significantly higher IFNγ responses than the Prevacent and PRRSGard groups, while the PrimePac group showed higher responses than PRRSGard. In contrast, no significant differences were observed among vaccinated groups following NC23-11 restimulation (Figure 4F). Overall, IFNγ responses to NC134, NC20-1, and NC23-11 were lower than those observed for the other strains. The homologous restimulation results for each vaccinated group are shown in Appendix A.

### 3.4. T Cell Epitope Coverage Between Vaccine Strains and NC PRRSV-2 Field Strains

In this study, the T cell epitope content of NC PRRSV-2 field strains shared with the vaccine strains was assessed using the PigMatrix tool [36] the EpiCC algorithm and expressed as T cell epitope coverage for both SLA class I and class II.

#### 3.4.1. T Cell Epitope Coverage of SLA Class I and Relationship Between T Cell Epitope Coverage and the Mean Frequency of PRRSV-2 Specific IFNγ–Producing Cells in Each Vaccinated Group

The results demonstrated that T cell epitope coverage of vaccine strains exhibited the dynamic patterns for each gene against different PRRSV-2 strains (Figure 5A–E). The reference strain, VR2332, showed higher epitope coverage across several structural genes, particularly in the Ingelvac group. Among the NC PRRSV-2 field strains, NC134 exhibited notably higher epitope coverage in the GP2b, M, and N genes in all vaccines. Furthermore, NC23-11 and NC24-9 displayed similar epitope coverage profiles across genes in all vaccines, consistent with their classification within the same PRRSV-2 lineage. Variable levels of T cell epitope coverage were observed in NC20-1 and NC24-6 (Figure 5A–E).

To assess T cell epitope coverage against NC PRRSV-2 field strains, the VR2332 reference strain was excluded, and the average SLA class I epitope coverage for each vaccine was calculated across the NC PRRSV-2 strains. The results showed that the average T cell epitope coverage of SLA class I across NC PRRSV-2 strains varied by gene and vaccine (Figure 5F). For Fostera, coverage ranged from 42.64% (GP3) to 65.51% (GP4); for Ingelvac, from 44.14% (GP3) to 62.80% (GP4); for PrimePac, from 42.47% (GP3) to 64.53% (M); for Prevacent, from 40.44% (GP2b) to 62.73% (M); and for PRRSGard, from 43.17% (GP3) to 72.48% (GP5a). Overall, the highest average T cell epitope coverage was observed in GP4 (60.82%), GP5a (60.37%), and M (59.06) genes, while GP3 (42.93%), GP2b (48.82%), and GP2a (51.38%) exhibited the lowest coverage across all vaccine groups (Figure 5F).

To evaluate vaccine immunogenicity, we assessed the correlation between T cell epitope coverage of epitopes identified for each gene and the relevant vaccine strain, and the mean frequency of PRRSV-2 specific IFNγ–producing cells in each vaccinated group after different PRRSV-2 restimulation. The results demonstrated a significant correlation (*p* < 0.05) between SLA class I T cell epitope coverage of the N gene and the mean number of PRRSV-specific IFNγ–producing cells (Figure 6A), while no significant correlations were found for the other genes or strains.

#### 3.4.2. T Cell Epitope Coverage of SLA Class II and Relationship Between T Cell Epitope Coverage and the Mean Frequency of PRRSV-2 Specific IFNγ–Producing Cells in Each Vaccinated Group

For SLA class II, the T cell epitope coverage patterns of vaccine strains mirrored those observed with SLA class I across structural genes when compared against various PRRSV-2 strains (Figure 7A–E). Consistent with SLA class I, the reference strain, VR2332, showed higher epitope coverage across several structural genes, particularly in the Ingelvac group. Among NC field strains, NC134 showed notably high coverage in GP2b, M, and N genes across all vaccines. For NC24-9, most vaccines had the highest epitope coverage in the GP4 gene, while the N gene showed the lowest, a pattern also seen in NC23-11. Additionally, NC20-1, NC23-11, and NC24-6 showed higher epitope coverage in the GP5a gene with Fostera, Ingelvac, and PRRSGard, and in the M gene with PrimePac and Prevacent. Conversely, the lowest epitope coverage varied by structural gene across vaccine groups (Figure 7A–E). The average SLA class II T cell epitope coverage for each vaccine was analyzed across the NC PRRSV-2 strains using the same method applied for SLA class I. The results showed that the average T cell epitope coverage of SLA class II across NC PRRSV-2 strains varied by gene and vaccine (Figure 7F). For Fostera, coverage ranged from 40.47% (GP2b) to 82.42% (GP5a); for Ingelvac, from 39.65% (GP2b) to 85.93% (GP5a); for PrimePac, from 37.92% (GP2a) to 68.36% (N); for Prevacent, from 38.72% (GP2a) to 63.08% (M); and for PRRSGard, from 36.64% (N) to 75.74% (GP5a). Overall, the highest average T cell epitope coverage was observed in GP5a (67.85%), GP4 (64.54%), and M (62.43%) genes, while GP2a (42.57%), GP3 (43.25%), and GP2b (45.03%) exhibited the lowest coverage across all vaccine groups (Figure 7F).

Similar to SLA class I, although the N gene did not exhibit the highest T cell epitope coverage against NC PRRSV-2 strains (Figure 7F), a significant correlation (*p* < 0.05) was observed between SLAII epitope coverage and mean PRRSV specific IFNγ–producing cells following VR2332 and NC24-6 restimulation (Figure 6B,C). No significant correlations were found for other genes or viral strains.

## 4. Discussion

Vaccination is widely employed to reduce the severity of clinical signs and the economic burden associated with PRRSV infection. However, an effective vaccine capable of inducing sterile immunity against both homologous and heterologous PRRSV strains has yet to be developed. Therefore, selecting appropriate commercial vaccines is a critical factor in controlling PRRSV infection. Notably, commercially available vaccines in the US are modified live vaccines, each belonging to distinct PRRSV-2 strain lineages. Despite the continuous emergence of new PRRSV-2 variants driven by high mutation rates and frequent recombination, new commercial vaccines have not been developed in response to newly emerging strains appearing over the years. This raises concerns about the ability of existing vaccines to prevent new outbreaks caused by emerging and re-emerging PRRSV variants. The objective of this study was to establish a PRRSV-2 immune bank using PBMCs and plasma collected from pigs vaccinated with commercially available vaccines in the US, followed by in vitro stimulation with circulating strains from the NC swine industry, to evaluate vaccine immunogenicity following vaccination. In addition, we compared T cell epitope coverage between vaccine and challenge strain, based on an earlier study that used Prevacent against challenge strains and determined that T cell content comparisons were well aligned with protective efficacy of the vaccine. These findings may provide valuable guidance and in vitro/in silico approaches for selecting appropriate vaccines against endemic and emerging PRRSV-2 strains.

Both nAbs and IFNγ production against homologous and heterologous PRRSV strains are key immune correlates of protection for predicting vaccine efficacy and immunogenicity [19,24]. The nAbs play a key role in viral clearance and protection against reinfection, as demonstrated by a study in which passive transfer of nAbs at a titer of 8 prevented viremia, while a titer of 32 conferred sterilizing immunity [17,19]. However, vaccine-induced neutralizing antibodies are slow to develop, inconsistently produced, and show limited cross-reactivity among PRRSV strains [19,20]. Interestingly, age influences the severity and susceptibility of PRRSV infection, with nursery pigs exhibiting greater vulnerability and experiencing more severe and prolonged infections compared to grower or adult pigs [37]. Moreover, wild-type PRRSV infections are most detected during the mid-growing phase of the pig [38]. The results highlight two key periods of susceptibility to PRRSV infection: the early nursery stage and the middle of the growing phase. In our study, pigs were vaccinated with different commercial vaccines after weaning, reflecting common US swine industry practices in which the majority of herds (90%) administer PRRSV modified-live vaccines at processing, weaning, or shortly thereafter [38]. In this study, all vaccinated groups exhibited humoral immune responses after vaccination. The results showed that none of the vaccinated groups produced nAbs against either the field PRRSV-2 strains or the reference strain at 4 weeks post-vaccination. This finding aligns with previous studies reporting undetectable nAbs at this time point [39,40,41,42,43]. Furthermore, when vaccinated pigs were challenged with the homologous strain at 4 weeks post-vaccination, they developed only minimal SN titers that failed to reach the protective threshold [44]. It indicated that nAbs may play a limited role in protecting against PRRSV during the early nursery stage but likely play a crucial role during the mid-growing phase after presence of nAbs. A recent study reported that some vaccinated pigs developed nAbs against both homologous and heterologous strains at 64 dpv [45]. This implies that timing post-vaccination plays a crucial role in the induction of nAbs in vaccinated pigs. Therefore, these findings highlight the limited ability of commercial PRRSV vaccines to induce nAbs against heterologous strains within a short period, highlighting the inadequacy of relying solely on nAbs production to predict vaccine immunogenicity and emphasizing the need for next-generation vaccine platforms.

Although nAbs production following vaccination is limited, the protection conferred by MLV vaccines appears to rely more on cellular immunity than on humoral immunity [46]. Several studies revealed PRRSV can persist in infected tissues despite the presence of nAbs [21,22] and can evade nAbs by spreading intercellularly through membrane nanotubes [47]. Notably, nAbs are generally strain-specific with limited cross-reactivity, while broad neutralization typically requires recognition of conformational epitopes [48]. Therefore, cell-mediated immune responses are likely to play a crucial role in viral clearance both before and after the emergence of nAbs. Moreover, cellular responses offer broader heterologous protection, while nAbs primarily confer homologous protection against PRRSV [19]. Evidence indicates that higher PRRSV-specific IFNγ–producing cells including CD4 T-cell response are associated with protection, as reflected by reduced viral loads or milder lung pathology following PRRSV challenge [24,49,50,51]. In this study, vaccine immunogenicity was assessed by measuring strain-specific immune responses previously identified as immune correlates of protection (CoP) in a prior study [19,24]. Following heterologous restimulation, the vaccinated groups exhibited varying levels of IFNγ–producing cells. This is consistent with a previous study, which reported that the number of IFNγ-producing cells in PBMCs varied after stimulation with heterologous viruses [52,53]. The variation in cellular immune responses among vaccinated groups exposed to different PRRSV-2 strains may reflect the unique immunological profiles of each strain, and such differences could help predict vaccine effectiveness against newly emerging strains. However, as our findings are based on in vitro experiments, further in vivo studies are required to validate the predictive capability of our immune biobank. In this study, T-cell subsets associated with IFNγ production were not identified in our immune biobank due to the limitations of the ELISPOT assay. To enhance the predictive capacity of future immune biobanks for cell-mediated immune profiling, combining flow cytometry with ELISPOT may improve the characterization of T cell–mediated responses by identifying specific T cell subsets involved in IFNγ production following vaccination. Interestingly, T-helper (Th) cell responses were the primary responders during viremia and were associated with a reduction in viral load, while TCR-γδ cells became active post-viremia [54]. Cytotoxic T lymphocytes (CTLs) responses were predominantly detected at the sites of infection, including the lung and bronchoalveolar lavage fluid [19]. According to Li et al., the predominant T cell subsets responsible for IFNγ production at 28 dpv and viral challenge were Th cells, followed by TCR-γδ cells, Th/memory cells, and CTLs [55]. However, other studies have demonstrated that Th/memory cells are often the dominant population after MLV vaccination or heterologous restimulation [56,57]. Notably, PRRSV-specific Th cells have been reported as the most reliable immune correlates of protection [24]. The variation in dominant T cell subset responses may be influenced by several factors, including the duration of vaccination, vaccine strains, PRRSV strains used for restimulation, and the age of the pigs. Therefore, tracking T cell subset immune responses weekly after vaccination may provide valuable insights into the cellular immune responses elicited by different PRRSV strains and vaccine groups.

In this study, sequence-based prediction using nucleotide and amino acid sequences of complete PRRSV genome was performed. However, there was no correlation between nucleotide and amino acid similarity of complete genome and the number of PRRSV-specific IFNγ producing cells in each PRRSV strain (). This finding aligns with previous studies showing that genetic similarity between vaccine and field strains is not a reliable predictor of the protective efficacy conferred by PRRSV MLV vaccines [58]. A recent study revealed that antigenic divergence and immune escape among PRRSV lineages are associated with T cell epitope diversity [59]. A high level of T cell epitope overlap between vaccine and challenge strains may contribute to stronger cross-reactive cell-mediated immune responses [28]. In the present study, we compared available commercial PRRSV-2 vaccines used in the U.S. with NC PRRSV-2 field strains, including a reference strain using T cell epitope content of eight structural proteins (GP2a, GP2b, GP3, GP4, GP5, GP5a, M, and N) with EpiCC algorithm.

The results demonstrate that vaccine strains and NC field strains exhibited varying degrees of T cell epitope coverage for both SLA class I and class II, likely contributing to differences in cell-mediated immune responses following restimulation with diverse PRRSV-2 strains. In our study, the highest T cell epitope coverage was primarily located in the GP4, GP5a, and M genes for both SLA classes. However, a recent study reported that among PRRSV-2 strains including NADC20, NADC30, and NC174, the N gene had the highest average vaccine epitope coverage, while the GP5 gene exhibited the lowest [28]. The differences in T cell epitope coverage may be attributed to the distinct PRRSV-2 strains used for analysis and prediction in our study. Our analysis found the positive correlations between T cell epitope coverages and IFNγ responses were significant for only N protein after restimulation with VR2332 (SLA class I and class II), and NC24-6 (SLA class II). This correlation is identified despite certain limitations, the major one being that while the EpiCC analysis is based on the individual structural PRRS proteins, in vitro results are obtained using full replication-competent PRRS virus. Therefore, the in vitro conditions are more complex than peptide-based assays. Stimulation with individual proteins or peptides might have revealed stronger correlations. However, the importance of certain structural proteins for protective immunity is noteworthy and consistent with the findings of a previous study suggesting that GP2b, GP2, M, and N are more closely associated with protective immunity [28]. Although the N gene did not exhibit the highest T cell epitope coverage compared to GP4, GP5a, and M genes, it may contain critical epitopes that synergize with epitopes from other genes to induce cross-protective immunity. Notably, the N protein can interact with its own GP5 and E proteins, as well as host proteins, influencing viral entry, replication, and the regulation of host signaling pathways, and it also contains multiple antigenic epitopes, including B and T cell epitopes, that are involved in the host immune response as extensively reviewed [60]. Previous studies have shown that certain regions of SLA Hp-4.0 haplotype-restricted CTL epitopes within the PRRSV M protein are capable of inducing PBMC proliferation and IFNγ production [61]. Taken together, our findings demonstrate substantial evolutionary differences between the MLV strains and the current NC circulating strains, with complete genome amino acid similarity below 61%. This was consistent with the low percentage of T cell epitope coverage across genes analyzed by EpiCC tool, underscoring the need to update commercial vaccines to keep pace with viral evolution. Moreover, predicting vaccine immunogenicity through T-cell epitope analysis using the EpiCC tool associated with protection may not only provide a valuable framework for vaccine selection but also guide the development of next-generation platforms, including peptide-based and T cell–targeted vaccines.

Our work represents an initial step toward developing methods for predicting vaccine immunogenicity using an immune biobank. Our findings suggest that the PRRSV immune biobank may be useful to predict vaccine immunogenicity by identifying which vaccine induces the strongest immune responses both humoral and cellular immunity against emerging PRRSV-2 strains. Nevertheless, we recognize the limitations of the current immune biobank. Future enhancements, such as extending blood sampling to later time points, evaluating cytokine responses both early and late after vaccination, and incorporating flow cytometry for T cell subset profiling, will strengthen the predictive capacity of our next-generation immune biobank. These would enhance the chances to determine heterologous neutralizing antibodies, including the prediction of vaccine immunogenicity based on the humoral immune system, together with the evaluation of specific T cell responses.

## 5. Conclusions

In conclusion, this study demonstrated that the PRRSV-2 immune biobank can serve as a useful tool for predicting vaccine immunogenicity based on immune correlates of protection using PBMCs and plasma against different PRRSV-2 strains. Although none of the vaccinated pigs developed neutralizing antibodies against NC PRRSV strains at 28 dpv, varying levels of IFNγ production upon restimulation suggest that cellular immunity may play an important role against different PRRSV-2 strains. The magnitude of these responses could help guide vaccine selection to better target specific PRRSV-2 strains. Additionally, this study highlights that predicting T cell epitope coverage of individual genes using the EpiCC algorithm between vaccine and NC PRRSV-2 strains can be used as complementary tool to enhance the predictive capacity of the immune biobank. In this study, we found a positive correlation between T cell epitope coverage of the N protein and the mean frequency of PRRSV-specific IFNγ–producing cells against VR2332 and NC24-6, supporting the potential role of T cell epitope coverage in heterologous cross-protection. These findings may guide future approaches for predicting vaccine immunogenicity; however, animal challenge models are needed to validate their predictive capacity for protective efficacy against diverse PRRSV-2 strains.

## Figures and Tables

**Figure 1 vaccines-13-01052-f001:**
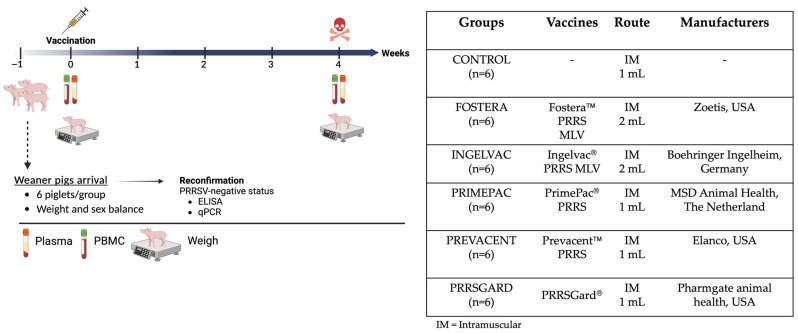
Experimental design for the establishment of a PRRSV-2 immune biobank. The schematic illustrates the study design from animal arrival to the experimental endpoint. Groups of pigs were vaccinated with different commercial PRRSV-2 vaccines at 0 dpv. As indicated in the timeline, PBMCs isolation and plasma collection were performed to assess the immune correlation of protection both humoral and cellular immune responses at 28 dpv. Created in BioRender. Sirisereewan, C. (2025) https://BioRender.com/2qdge6b, 8 September 2025.

**Figure 2 vaccines-13-01052-f002:**
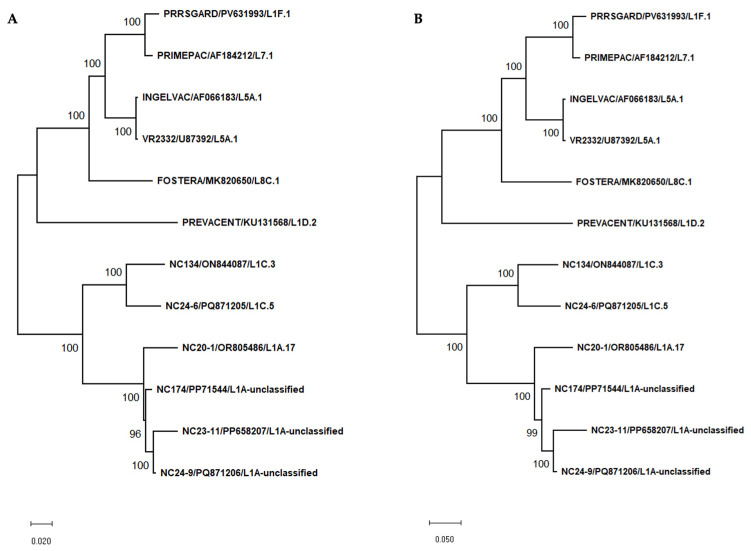
Phylogenetic tree based on complete genome sequences (**A**) and complete amino acid sequences (**B**) of vaccine strains and various PRRSV-2 strains used in this study. Phylogenetic trees were constructed using the Maximum Likelihood method with 1000 bootstrap replicates. The GTR+G substitution model was applied for nucleotide sequences, and the JTT+G model was used for amino acid sequences.

**Figure 3 vaccines-13-01052-f003:**
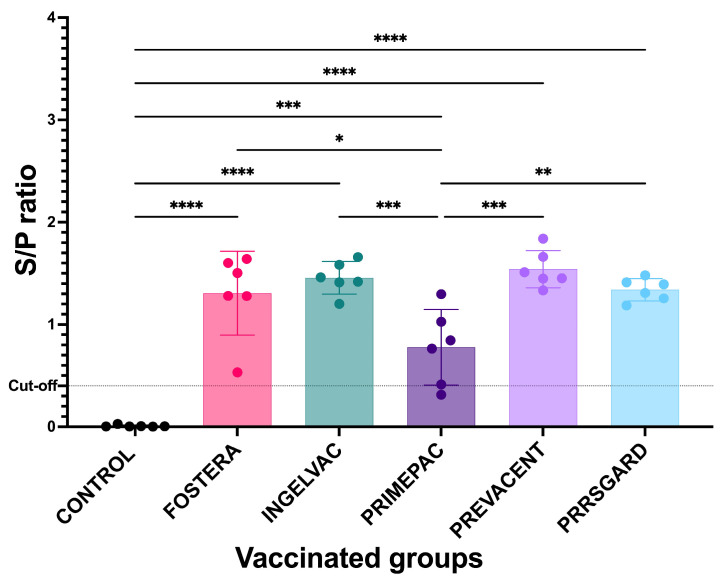
PRRSV-specific antibodies from vaccinated and control groups at 28 dpv. All data are presented as mean ± SD. Statistical analyses were performed using one-way ANOVA followed by Tukey’s multiple comparison test. Asterisks indicate statistically significant differences between groups (* *p* < 0.05, ** *p* < 0.01, *** *p* < 0.001, **** *p* < 0.0001). The dotted line indicates the positive cut off set at 0.4.

**Figure 4 vaccines-13-01052-f004:**
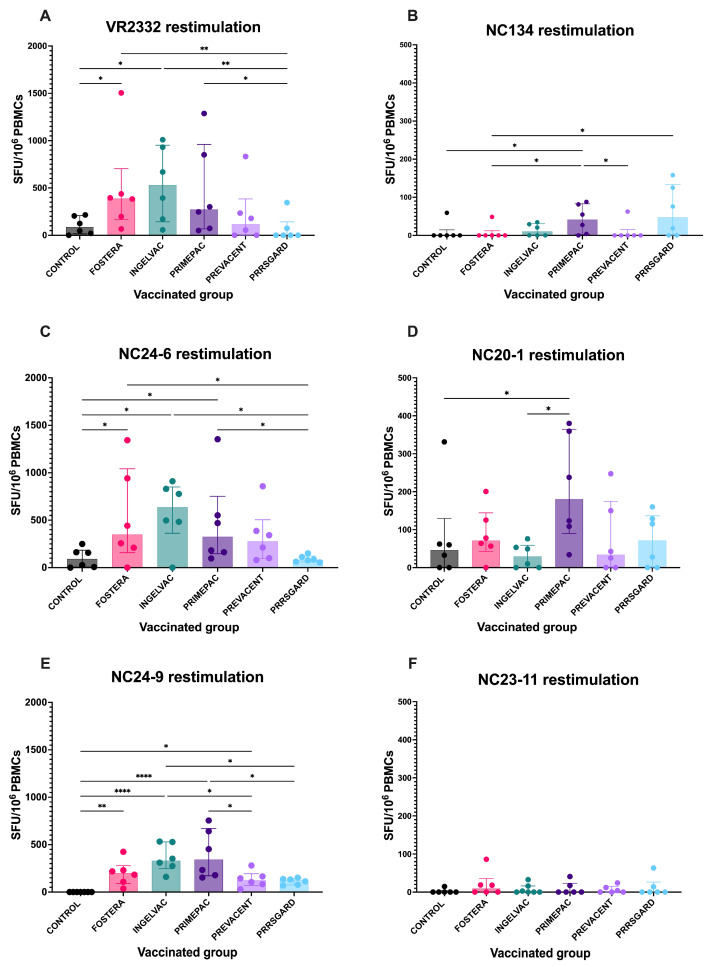
PRRSV-specific IFNγ–producing cells following restimulation with different PRRSV-2 strains in vaccinated groups. PBMCs collected at 28 dpv were stimulated with six PRRSV-2 strains: (**A**) VR2332, (**B**) NC134, (**C**) NC24-6, (**D**) NC20-1, (**E**) NC24-9, and (**F**) NC23-11. The number of IFNγ–producing cells were measured using ELISpot assay and presented as spot-forming units (SFU) per 10^6^ PBMCs. Each bar represents the median ± interquartile range for each vaccinated group (Fostera, Ingelvac, PrimePac, Prevacent, PRRSGard) compared with unvaccinated controls. Statistical analyses were performed using Kruskal–Wallis test followed by uncorrected Dunn’s multiple comparisons test. Asterisks indicate statistically significant differences between groups (* *p* < 0.05, ** *p* < 0.01, **** *p* < 0.0001).

**Figure 5 vaccines-13-01052-f005:**
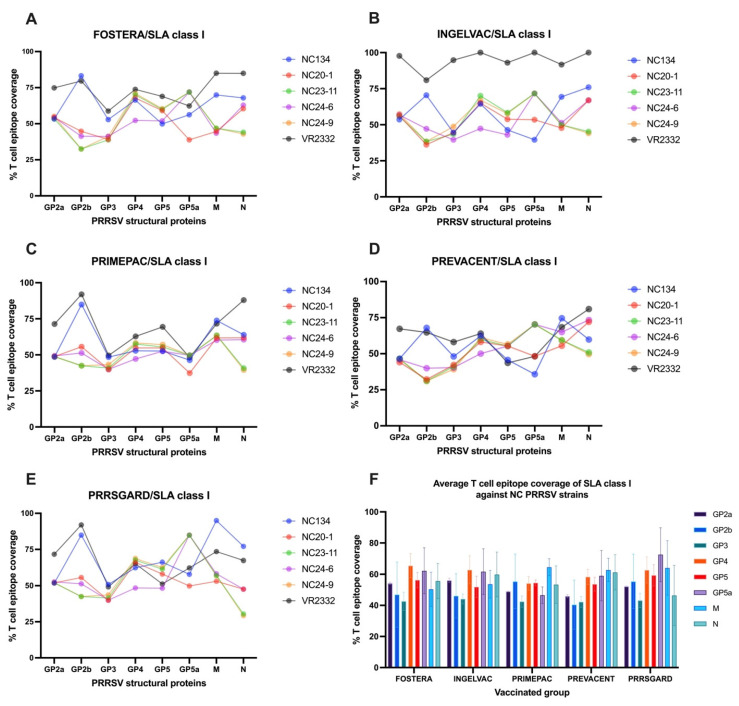
T cell epitope coverage scores for SLA class I are shown for eight PRRSV-2 structural proteins, comparing commercial vaccines with different PRRSV-2 strains: (**A**) Fostera, (**B**) Ingelvac, (**C**) PrimePac, (**D**) Prevacent, and (**E**) PRRSGard. (**F**) The average SLA class I T cell epitope coverage percentages against NC PRRSV-2 strains are also presented.

**Figure 6 vaccines-13-01052-f006:**
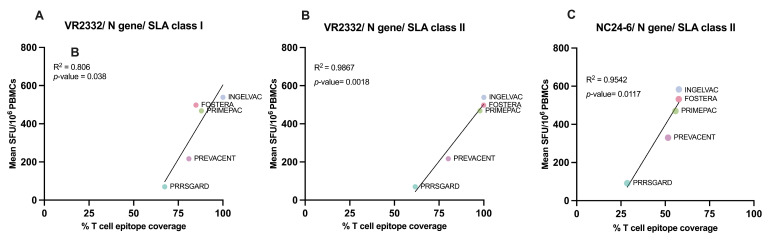
Correlation between predicted T cell epitope coverage scores and IFNγ responses in vaccinated groups. Scatter plots show the relationship between predicted N gene T cell epitope coverage (%) and mean IFNγ ELISpot responses (SFU/10^6^ PBMCs) for commercial vaccines. (**A**) VR2332, SLA class I; (**B**) VR2332, SLA class II; (**C**) NC24-6, SLA class II. All comparisons showed positive correlations, with R^2^ and *p*-values indicated on each plot.

**Figure 7 vaccines-13-01052-f007:**
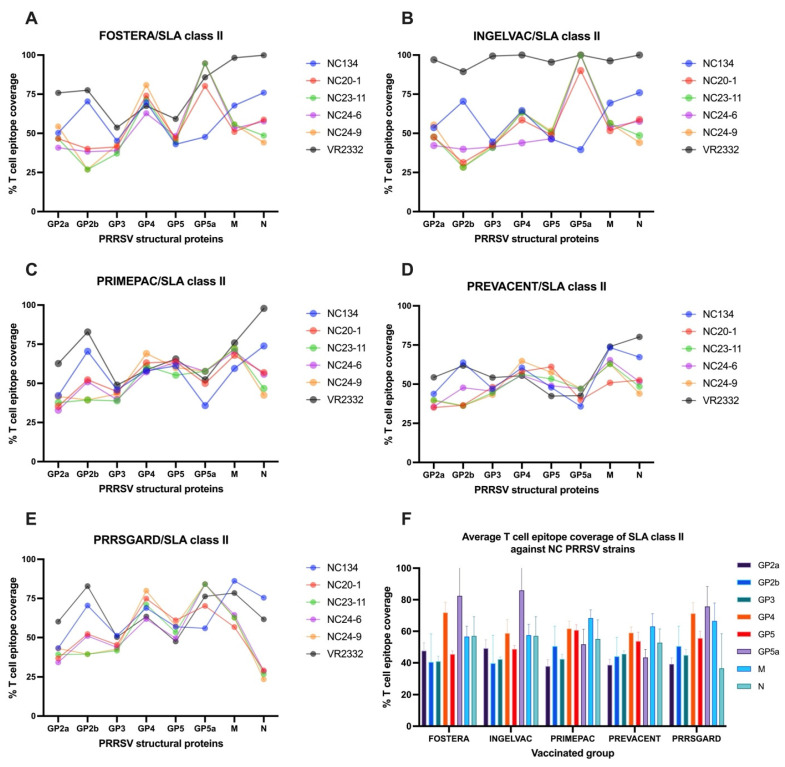
T cell epitope coverage scores for SLA class II are shown for eight PRRSV structural proteins, comparing commercial vaccines with different PRRSV-2 strains: (**A**) Fostera, (**B**) Ingelvac, (**C**) PrimePac, (**D**) Prevacent, and (**E**) PRRSGard. (**F**) The average SLA class II T cell epitope coverage percentages against NC PRRSV-2 strains are also presented.

## Data Availability

The original contributions presented in this study are included in the article/Appendix A. Further inquiries can be directed to the corresponding author.

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
