# Peer review of "Establishment of Immune Biobank for Vaccine Immunogenicity Prediction Using In Vitro and In Silico Methods Against Porcine Reproductive and Respiratory Syndrome Virus"

_vaccines, 2025, doi:10.3390/vaccines13101052_

Round 1

Reviewer 1 Report

Comments and Suggestions for Authors

 Establishment of immune biobank for vaccine immunogenicity prediction using in vitro and in silico methods against porcine reproductive and respiratory syndrome virus

The overall goal of this study is intriguing and highly relevant for veterinarians, as prediction of vaccine efficacy is critical when a vaccine must be selected. Importantly, the inclusion of multiple vaccines and PRRSV strains—including recent isolates—allows the authors to draw robust and meaningful conclusions.

Major Comments

  1. Previous studies have demonstrated that virus-neutralizing (VN) antibody responses to PRRSV vaccination often appear late, whereas other immune markers such as cytokines can be detected within the first week post-vaccination. Taken together, these findings suggest that both early (3–7 days post-immunization) and late (around 42 dpi) sampling points are critical for biobank collection. Including this consideration in the study design—or at least acknowledging it as a limitation of the current biobank—would strengthen the manuscript.
  2. If this biobank is intended to serve as a predictive model for immune responses, the inclusion of clearly defined “positive” and “negative” response groups would be valuable. For instance, a prototype vaccine based on one of the PRRSV strains included in the study could represent a “best-case” scenario, i.e., 100% homologous immunity, whereas a highly heterologous European strain could serve as a “worst-case” scenario. Vaccine responses would then fall within this range.
  3. Given the high genetic similarity between the Ingelvac vaccine and VR2332, the Ingelvac/VR2332 combination could serve as the homologous reference. Since some level of neutralizing antibody response would be expected under these conditions, I recommend that the authors analyze and/or discuss their data accordingly.
  4. Beyond the general discussion of VN antibodies in PRRSV protection, it is essential to clarify whether the VN assay was conducted under optimal conditions. For example, do the authors believe that using 100 TCID₅₀ would have improved assay sensitivity? This issue is particularly important, as the study aims to establish a standardized biobank for future research. The absence of detectable nAbs after vaccination may therefore reflect a limitation of the biobank design rather than either (i) an inherent inability of the vaccines to induce VN responses or (ii) the limited role of VN antibodies as a correlate of protection.
  5. In the introduction, the rationale for selecting the specific correlates of protection (COPs) should be described in greater detail. A clearer justification would help readers understand why these parameters were prioritized over others and how they align with the overall goals of establishing the biobank.

Author Response

Reviewer #1

The overall goal of this study is intriguing and highly relevant for veterinarians, as prediction of vaccine efficacy is critical when a vaccine must be selected. Importantly, the inclusion of multiple vaccines and PRRSV strains—including recent isolates—allows the authors to draw robust and meaningful conclusions.

Major Comments

  1. Previous studies have demonstrated that virus-neutralizing (VN) antibody responses to PRRSV vaccination often appear late, whereas other immune markers such as cytokines can be detected within the first week post-vaccination. Taken together, these findings suggest that both early (3–7 days post-immunization) and late (around 42 dpi) sampling points are critical for biobank collection. Including this consideration in the study design—or at least acknowledging it as a limitation of the current biobank—would strengthen the manuscript.

Response 1: Thank you for pointing this out. We acknowledged the limitation of the current biobank you have suggested. We are planning to do a new biobank (2.0) that will take into account the current limitations and will consider the reviewer’s comments.  The new study design will include detection of cytokines during the early and late immune responses following vaccination, and the evaluation of neutralizing antibodies and cellular immunity later time points (35-42 dpv). Additionally, the new biobank will include a challenge with a PRRSV-2 strain currently circulating in NC. For better clarify, we added the sentence to explain our limitation on page 16, line 516-520 “Nevertheless, we recognize the limitations of the current immune biobank. Future enhancements, such as extending blood sampling to later time points, evaluating cytokine responses both early and late after vaccination, and incorporating flow cytometry for T cell subset profiling, will strengthen the predictive capacity of our next-generation immune biobank.”

  1. If this biobank is intended to serve as a predictive model for immune responses, the inclusion of clearly defined “positive” and “negative” response groups would be valuable. For instance, a prototype vaccine based on one of the PRRSV strains included in the study could represent a “best-case” scenario, i.e., 100% homologous immunity, whereas a highly heterologous European strain could serve as a “worst-case” scenario. Vaccine responses would then fall within this range.

Response 2: We thank the reviewer for this constructive suggestion. We agree that including well-defined “benchmark” groups representing best- and worst-case vaccine responses would improve the interpretability of a predictive biobank. However, our current study represents an initial step toward predicting vaccine immunogenicity, and genetic similarity between vaccine and field strains has proven to be an unreliable predictor of protective immunity. Therefore, an animal challenge model is required to validate predictive models of immune responses that correlate with protection or decreased disease severity after challenge. We are currently planning “Biobank 2.0,” that will include a challenge with a circulating NC PRRSV-2 strain. Additionally, at the moment we don’t have any available PRRSV-1 strain that can be used as “worst-case” scenario, but we have tested each vaccine group toward their homologous vaccine strains (supplementary material). For the next biobank we will be working to increase the NC strains available and to isolate an NC PRRSV-1 strain.

  1. Given the high genetic similarity between the Ingelvac vaccine and VR2332, the Ingelvac/VR2332 combination could serve as the homologous reference. Since some level of neutralizing antibody response would be expected under these conditions, I recommend that the authors analyze and/or discuss their data accordingly.

Response 3: We thank the reviewer for this important point. In our study, we designated VR-2332 as the reference strain. Although the Ingelvac vaccine and VR-2332 share high genetic similarity, this does not guarantee the induction of neutralizing antibodies (nAbs), particularly at early time points due to the characteristically delayed nAb response for PRRSV.

This is supported by previous research. For example, a study showed that pigs vaccinated with the Ingelvac vaccine and challenged with VR-2332 at 4 weeks post-vaccination developed only minimal SN titers. These titers were not significantly different from those in unvaccinated pigs and did not reach the cut-off level considered protective.

Li, Y.; Xu, L.; Jiao, D.; Zheng, Z.; Chen, Z.; Jing, Y.; Li, Z.; Ma, Z.; Feng, Y.; Guo, X.; et al. Genomic similarity and antibody-dependent enhancement of immune serum potentially affect the protective efficacy of commercial MLV vaccines against NADC30-like PRRSV. Virol Sin 2023, 38, 813-826, doi:10.1016/j.virs.2023.08.010.

We added the sentences and reference in discussion on page 14, line 409-411 “Furthermore, when vaccinated pigs were challenged with the homologous strain at 4 weeks post-vaccination, they developed only minimal SN titers that failed to reach the protective threshold”.

  1. Beyond the general discussion of VN antibodies in PRRSV protection, it is essential to clarify whether the VN assay was conducted under optimal conditions. For example, do the authors believe that using 100 TCID₅₀ would have improved assay sensitivity? This issue is particularly important, as the study aims to establish a standardized biobank for future research. The absence of detectable nAbs after vaccination may therefore reflect a limitation of the biobank design rather than either (i) an inherent inability of the vaccines to induce VN responses or (ii) the limited role of VN antibodies as a correlate of protection.

Response 4: We thank the reviewer for this excellent point. We fully agree that using 100 TCID₅₀ is a widely adopted approach and can improve sensitivity. However, in this study, we used 200 TCID₅₀ to assess potent responses, with the intention of capturing strong neutralizing antibody activity capable of overcoming a more stringent viral challenge, as may occur during natural infection.

We believe that the virus concentration used in our neutralization assay falls within the acceptable range, as several PRRSV neutralization studies have employed viral doses between 100 and 200 TCID₅₀.

100 TCID50

  • Yoon, K.J.; Zimmerman, J.J.; Swenson, S.L.; McGinley, M.J.; Eernisse, K.A.; Brevik, A.; Rhinehart, L.L.; Frey, M.L.; Hill, H.T.; Platt, K.B. Characterization of the humoral immune response to porcine reproductive and respiratory syndrome (PRRS) virus infection. J Vet Diagn Invest 1995, 7, 305-312, doi:10.1177/104063879500700302.
  • Martinez-Lobo, F.J.; Diez-Fuertes, F.; Simarro, I.; Castro, J.M.; Prieto, C. The Ability of Porcine Reproductive and Respiratory Syndrome Virus Isolates to Induce Broadly Reactive Neutralizing Antibodies Correlates With In Vivo Protection. Front Immunol 2021, 12, 691145, doi:10.3389/fimmu.2021.691145.

200 TCID50

  • Trible, B.R.; Popescu, L.N.; Monday, N.; Calvert, J.G.; Rowland, R.R. A single amino acid deletion in the matrix protein of porcine reproductive and respiratory syndrome virus confers resistance to a polyclonal swine antibody with broadly neutralizing activity. J Virol 2015, 89, 6515-6520, doi:10.1128/JVI.03287-14.
  • Li, Y.; Xu, L.; Jiao, D.; Zheng, Z.; Chen, Z.; Jing, Y.; Li, Z.; Ma, Z.; Feng, Y.; Guo, X.; et al. Genomic similarity and antibody-dependent enhancement of immune serum potentially affect the protective efficacy of commercial MLV vaccines against NADC30-like PRRSV. Virol Sin 2023, 38, 813-826, doi:10.1016/j.virs.2023.08.010.

In general, MLV vaccination showed a delayed neutralizing antibody response. We have acknowledged this limitation in our discussion (Page 14, Lines 415-416), where we state: "This implies that timing post-vaccination plays a crucial role in the induction of nAbs in vaccinated pigs.” Furthermore, motivated by this finding and the reviewer's valuable suggestion, our planned future studies will involve extending blood sampling to later time points to fully assess nAb kinetics for all the vaccines

  1. In the introduction, the rationale for selecting the specific correlates of protection (COPs) should be described in greater detail. A clearer justification would help readers understand why these parameters were prioritized over others and how they align with the overall goals of establishing the biobank.

Response 5: We thank the reviewer for this constructive suggestion. For better clarify, we added the sentences in the introduction including the references on page 2, line 77-82 “CoP was defined as the measurable immunological characteristics and mechanisms associated with protection from disease and responsible for vaccine-induced efficacy. In this study, we focused on two key arms of the CoP against PRRSV: nAbs, which serve as a humoral shield to prevent infection, and cell-mediated immunity, which is critical for broad and effective viral clearance. Both have been identified as relevant CoP in the PRRSV literature.”

Plotkin, S.A. Immunologic correlates of protection induced by vaccination. Pediatr Infect Dis J 2001, 20, 63-75, doi:10.1097/00006454-200101000-00013.

Plotkin, S.A. Correlates of protection induced by vaccination. Clin Vaccine Immunol 2010, 17, 1055-1065, doi:10.1128/CVI.00131-10.

Plotkin, S.A. Recent updates on correlates of vaccine-induced protection. Front Immunol 2022, 13, 1081107, doi:10.3389/fimmu.2022.1081107.

Kick, A.R.; Grete, A.F.; Crisci, E.; Almond, G.W.; Kaser, T. Testable Candidate Immune Correlates of Protection for Porcine Reproductive and Respiratory Syndrome Virus Vaccination. Vaccines (Basel) 2023, 11, doi:10.3390/vaccines11030594.

Reviewer 2 Report

Comments and Suggestions for Authors

This manuscript presents a well-conceived and executed study addressing a critical challenge in porcine health: predicting the immunogenicity of commercial PRRSV-2 vaccines against heterologous, circulating strains. The authors successfully establish an immune bank and employ a combination of sophisticated in vitro (ELISPOT, serology) and in silico (EpiCC algorithm) methods to evaluate vaccine-induced responses. The work is novel and provides a valuable framework for a more rational selection of vaccines in the field. The manuscript is generally well-written, the experiments are appropriately designed, and the data support the conclusions. I recommend publication after minor revisions to address the points below.

1. The absence of detectable nAbs at 28 dpv is presented, but this time point is likely too early to draw definitive conclusions about the humoral arm of protection. PRRSV nAbs are known to develop slowly.

2.The EpiCC tool analyzes individual proteins, while the ELISPOT uses whole virus for re-stimulation. This discrepancy might explain why stronger correlations were not found across more proteins/strains.

3. The study is entirely based on in vitro and in silico correlates. While these are valuable predictive tools, the ultimate validation is protection against a live virus challenge.

4. The manuscript reports the use of the EpiCC algorithm to predict Tcell epitopes;however, the provenance and access details of this tool are not specified. Please cite the original publication(s) describing EpiCC and provide complete source information—including the developer or maintaining institution, version and release date, and an access link or DOI—as well as the website or repository from which the tool can be obtained. If the platform is proprietary, please indicate the commercial provider and licensing terms. This information is necessary to enable independent evaluation and reproducibility of the analyses.

5. Figure 4 contains a large number of comparisons. While statistically sound, it is somewhat difficult to parse.

6. Table 1 described in manuscript should be Table S1.

Author Response

Reviewer #2

This manuscript presents a well-conceived and executed study addressing a critical challenge in porcine health: predicting the immunogenicity of commercial PRRSV-2 vaccines against heterologous, circulating strains. The authors successfully establish an immune bank and employ a combination of sophisticated in vitro (ELISPOT, serology) and in silico (EpiCC algorithm) methods to evaluate vaccine-induced responses. The work is novel and provides a valuable framework for a more rational selection of vaccines in the field. The manuscript is generally well-written, the experiments are appropriately designed, and the data support the conclusions. I recommend publication after minor revisions to address the points below.

  1. The absence of detectable nAbs at 28 dpv is presented, but this time point is likely too early to draw definitive conclusions about the humoral arm of protection. PRRSV nAbs are known to develop slowly.

Response 1: We thank the reviewer for this excellent point. We agree that 28 days post-vaccination is likely too early to draw definitive conclusions about nAbs induction, given that PRRSV nAbs are known to develop slowly. We have acknowledged this limitation in our discussion (Page 14, Lines 415-416), where we state: "This implies that timing post-vaccination plays a crucial role in the induction of nAbs in vaccinated pigs.” Furthermore, motivated by this finding and the reviewer's valuable suggestion, our planned future studies will involve extending blood sampling to later time points to fully assess nAb kinetics.

  1. The EpiCC tool analyzes individual proteins, while the ELISPOT uses whole virus for re-stimulation. This discrepancy might explain why stronger correlations were not found across more proteins/strains.

Response 2: Thank you for pointing out. This point is addressed in the discussion section of the manuscript on page 15-16, line 486-491 “This correlation is identified despite certain limitations, the major one being that while the EpiCC analysis is based on the individual structural PRRS proteins, in vitro results are obtained using full replication-competent PRRS virus. Therefore, the in vitro conditions are more complex than peptide-based assays and include innate immune responses in addition to adaptive responses. Stimulation with individual proteins or peptides might have revealed stronger correlations.”

  1. The study is entirely based on in vitro and in silico correlates. While these are valuable predictive tools, the ultimate validation is protection against a live virus challenge.

Response 3: We thank the reviewer for this excellent point. We agree that an in vivo challenge is the ultimate validation for both vaccine efficacy and our predictive models. Our current study represents a crucial first step in developing these tools, and we are planning future experiments that will include a live virus challenge with a circulating NC PRRSV-2 strain to help correlate in vivo and in vitro data.

  1. The manuscript reports the use of the EpiCC algorithm to predict Tcell epitopes; however, the provenance and access details of this tool are not specified. Please cite the original publication(s) describing EpiCC and provide complete source information—including the developer or maintaining institution, version and release date, and an access link or DOI—as well as the website or repository from which the tool can be obtained. If the platform is proprietary, please indicate the commercial provider and licensing terms. This information is necessary to enable independent evaluation and reproducibility of the analyses.

Response 4: We thank the reviewer for this constructive suggestion. We have now added the appropriate citations for the PigMatrix tool and the EpiCC algorithm. Specifically, the reference for the PigMatrix tool on page 10, line 297  has been corrected, and the reference for the EpiCC algorithm has been added on page 2, line 83, and on page 5, line 197.

The full citations are:

For PigMatrix:

Gutiérrez, A.H. et al. (2015) “Development and validation of an epitope prediction tool for swine (PigMatrix) based on the pocket profile method,” BMC Bioinformatics, 16(1), p. 290. Available at: https://doi.org/10.1186/s12859-015-0724-8.

For EpiCC algorithm:

Gutiérrez, A.H. et al. (2017) “T‐cell epitope content comparison (EpiCC) of swine H1 influenza A virus hemagglutinin,” Influenza and Other Respiratory Viruses, 11(6), pp. 531–542. Available at: https://doi.org/10.1111/irv.12513.

  1. Figure 4 contains a large number of comparisons. While statistically sound, it is somewhat difficult to parse.

Response 5: We thank the reviewer for this constructive feedback and acknowledge that Figure 4 is dense due to the large number of comparisons. However, we believe that presenting all vaccinated groups together in a single figure is essential for a direct and comprehensive comparison of their respective IFN-γ responses. Splitting the data across multiple figures would make it more difficult for the reader to appreciate the full spectrum and relative magnitude of the responses between groups.

  1. Table 1 described in manuscript should be Table S1.

Response 6: Thank you for identifying this mistake. We replaced the Table1 to Table S1 on page 4, line 114.

Reviewer 3 Report

Comments and Suggestions for Authors

In this study, the authors established a PRRSV-2 immune bank using PBMCs and plasma collected from pigs vaccinated with commercially available vaccines in the US, followed by in vitro stimulation with relevant strains, to evaluate the vaccine immunogenicity following vaccination. The authors raise a very important question: which vaccine strains are most effective in mitigating outbreaks caused by newly emerging PRRSV strains? The article is interesting and worthy of publication. 
This study has a major limitation: the lack of in vivo confirmation of the results. The authors should have infected several immunized animals and assessed protection against at least several new strains. However, this would have significantly increased the number of animals and perhaps the authors did not have this opportunity. Although the authors mention this in their conclusions, they should more clearly state that this is a limitation of the study.
In my opinion, the authors stated different study objectives in the Introduction and the Discussion sections. The introduction states that the study's goal is to assess vaccine immunogenicity, including both humoral and cell-mediated responses, and to identify commercial PRRSV vaccines that provide strong protection against various PRRSV-2 strains. However, the obtained results do not allow one to assess which commercial vaccines are effective against new strains. The authors should be more cautious and not claim that the study results allow predicting vaccine immunogenicity and guiding vaccine selection against newly emerging PRRSV strains in endemic regions.
While the authors have indeed obtained interesting results, this is still a first step toward developing a system for in vitro evaluation of PRRS vaccine efficacy. Therefore, the authors should make some minor adjustments to the manuscript text to avoid misleading readers.

Author Response

 Review #3

In this study, the authors established a PRRSV-2 immune bank using PBMCs and plasma collected from pigs vaccinated with commercially available vaccines in the US, followed by in vitro stimulation with relevant strains, to evaluate the vaccine immunogenicity following vaccination. The authors raise a very important question: which vaccine strains are most effective in mitigating outbreaks caused by newly emerging PRRSV strains? The article is interesting and worthy of publication. 
This study has a major limitation: the lack of in vivo confirmation of the results. The authors should have infected several immunized animals and assessed protection against at least several new strains. However, this would have significantly increased the number of animals and perhaps the authors did not have this opportunity. Although the authors mention this in their conclusions, they should more clearly state that this is a limitation of the study.
In my opinion, the authors stated different study objectives in the Introduction and the Discussion sections. The introduction states that the study's goal is to assess vaccine immunogenicity, including both humoral and cell-mediated responses, and to identify commercial PRRSV vaccines that provide strong protection against various PRRSV-2 strains. However, the obtained results do not allow one to assess which commercial vaccines are effective against new strains. The authors should be more cautious and not claim that the study results allow predicting vaccine immunogenicity and guiding vaccine selection against newly emerging PRRSV strains in endemic regions.
While the authors have indeed obtained interesting results, this is still a first step toward developing a system for in vitro evaluation of PRRS vaccine efficacy. Therefore, the authors should make some minor adjustments to the manuscript text to avoid misleading readers.

We thank the reviewer for this excellent point. We agree that an in vivo challenge is the ultimate validation for both vaccine efficacy and our predictive models. Our current study represents a crucial first step in developing these tools, and we are planning future experiments that will include a live virus challenge with a circulating NC PRRSV-2 strain to help correlate in vivo and in vitro data.

In our manuscript we have stated vaccine “immunogenicity” because we are aware that this system is not predicting vaccine “efficacy”, therefore we hope this term is not misleading for readers.

Reviewer 4 Report

Comments and Suggestions for Authors

The authors developed a biobank of PRRSV-vaccine exposed porcine immune cells and tried to predict vaccine efficacy in vitro. Although the idea is interesting, there are a number of points that require clarification or expansion to improve the robustness, accuracy, and clarity of the manuscript. Specific comments follow.

Major points:

  1. Please justify to immunize animals just once but not more to induce nAbs.
  2. Line 62: Please explain the difference between PRRSV and PRRSV-2.
  3. Line 108: I don’t see Table 1.
  4. Line 110: Please describe the source of PAMs and VR2332.
  5. Line 165: Please explain “complete media”.
  6. Figure 3: Please include amino acid homologies of each vaccine against ELISA antigen strain in the graph to help understand the difference of antibody levels.

Minor points:

  1. Figure 1: “-7” should be “-1” or change all numbers in days.

Author Response

Reviewer #4

The authors developed a biobank of PRRSV-vaccine exposed porcine immune cells and tried to predict vaccine efficacy in vitro. Although the idea is interesting, there are a number of points that require clarification or expansion to improve the robustness, accuracy, and clarity of the manuscript. Specific comments follow.

Major points:

  1. Please justify to immunize animals just once but not more to induce nAbs.

Response 1:  We thank the reviewer for this point. While a single vaccination is standard according to manufacturer’s instructions, the delayed development of neutralizing antibodies is a well-documented limitation. This presents a challenge: a homologous booster might improve strain-specific efficacy but would not provide the broad, cross-protective immunity needed against diverse field strains.

Furthermore, using different PRRSV MLV vaccines as a booster is not practical in the field, as this could introduce new viral strains into a herd, causing more problems than benefits. Therefore, this study was designed to mimic field conditions by using a single vaccination in weaners to assess nAb production and vaccine immunogenicity within the recommended timeline.

  1. Line 62: Please explain the difference between PRRSV and PRRSV-2.

Response 2: We thank the reviewer for this important point. As PRRSV comprises two distinct species (PRRSV-1 and PRRSV-2). We have amended the text to 'commercial PRRSV-2 vaccines' (Page 2, Line 61) to clarify that our investigation was limited to the PRRSV-2 species and that all vaccines used were derived from PRRSV-2 strains.

  1. Line 108: I don’t see Table 1.

Response 3: Thank you for pointing out. We deleted the word Table 1 in the manuscript.

  1. Line 110: Please describe the source of PAMs and VR2332.

Response 4: Thank you for your inquiry. The PAMs used in this study were isolated from lungs of PRRSV-negative pigs after euthanasia. Those pigs were part of unrelated research projects or were used for teaching purposes. The VR2332 virus was the ATCC® VR-2332™ reference strain (GenBank: U87392), which was purchased from the American Type Culture Collection (ATCC). To better clarify, we added the sentence “The reference strain, ATCC® VR-2332™ (GenBank: U87392), was propagated in MA104 cell line.” on page 3, line 116-117.

  1. Line 165: Please explain “complete media”.

Response 5: Thank you for pointing out. The definition of complete media was explained in the sentences on page 5, line 165-169 “After isolation, the cells were resuspended in RPMI 1640 medium (GIBCO, Carlsbad, CA, USA), supplemented with 10% heat-inactivated, fetal bovine serum (FBS), (GIBCO), 100 U/mL of penicillin G, 100 μg/mL of streptomycin, 0.01 mg/mL of gentamycin (GIBCO), and 2 mM L-glutamine (GIBCO) referred as complete media.

  1. Figure 3: Please include amino acid homologies of each vaccine against ELISA antigen strain in the graph to help understand the difference of antibody levels.

Response 6: We thank the reviewer for this valuable suggestion. We are unable to provide a direct comparison of amino acid homologies because the study utilized a commercial ELISA kit, and the manufacturer does not disclose the specific identity of the coated PRRSV antigen. However, the antigen is known to be the PRRSV nucleocapsid (N) protein, which is highly conserved among PRRSV strains.

Minor points:

  1. Figure 1: “-7” should be “-1” or change all numbers in days.

Response 1: Thank you for identifying this mistake. We have corrected the number in Figure 1 from -7 to -1 accordingly.
